# Physical Comorbidities and Depression in Recent and Long-Term Adult Cancer Survivors: NHANES 2007–2018

**DOI:** 10.3390/cancers13133368

**Published:** 2021-07-05

**Authors:** Dafina Petrova, Andrés Catena, Miguel Rodríguez-Barranco, Daniel Redondo-Sánchez, Eloísa Bayo-Lozano, Rocio Garcia-Retamero, José-Juan Jiménez-Moleón, María-José Sánchez

**Affiliations:** 1CIBER of Epidemiology and Public Health (CIBERESP), 28029 Madrid, Spain; miguel.rodriguez.barranco.easp@juntadeandalucia.es (M.R.-B.); daniel.redondo.easp@juntadeandalucia.es (D.R.-S.); jjmoleon@ugr.es (J.-J.J.-M.); mariajose.sanchez.easp@juntadeandalucia.es (M.-J.S.); 2Escuela Andaluza de Salud Pública, 18080 Granada, Spain; 3Instituto de Investigación Biosanitaria ibs.GRANADA, 18012 Granada, Spain; 4Mind, Brain and Behavior Research Center, University of Granada, 18071 Granada, Spain; acatena@ugr.es (A.C.); rretamer@ugr.es (R.G.-R.); 5Department of Radiation Oncology, University Hospital Virgen Macarena, 49009 Seville, Spain; eloisa.bayo.sspa@juntadeandalucia.es; 6Harding Center for Risk Literacy, 14195 Berlin, Germany; 7Department of Preventive Medicine and Public Health, University of Granada, 18071 Granada, Spain

**Keywords:** cancer, comorbidity, mental health, depression, cancer survivors

## Abstract

**Simple Summary:**

Most cancer patients suffer one or more physical comorbidities (other somatic diseases present at the moment of cancer diagnosis). Previous research has shown that these comorbidities can interfere with cancer treatment and shorten the patient’s survival time. We propose that comorbidities could also interfere with the mental health of cancer patients and increase the risk of suffering depression in the years following the cancer diagnosis. We tested this possibility in a study of 2073 adult cancer survivors. We found that the number of physical comorbidities present at the moment of cancer diagnosis was related to higher risk of reporting depression in cancer survivors who were diagnosed up to 5 years before the study. This relationship was strongest among survivors of breast cancer. Information about comorbidities is usually readily available and could be useful in streamlining depression screening or targeting prevention efforts in cancer patients and survivors.

**Abstract:**

Many adult cancer patients present one or more physical comorbidities. Besides interfering with treatment and prognosis, physical comorbidities could also increase the already heightened psychological risk of cancer patients. To test this possibility, we investigated the relationship between physical comorbidities with depression symptoms in a sample of 2073 adult cancer survivors drawn from the nationally representative National Health and Nutrition Examination Survey (NHANES) (2007–2018) in the U.S. Based on information regarding 16 chronic conditions, the number of comorbidities diagnosed before and after the cancer diagnosis was calculated. The number of comorbidities present at the moment of cancer diagnosis was significantly related to depression risk in recent but not in long-term survivors. Recent survivors who suffered multimorbidity had 3.48 (95% CI 1.26–9.55) times the odds of reporting significant depressive symptoms up to 5 years after the cancer diagnosis. The effect of comorbidities was strongest among survivors of breast cancer. The comorbidities with strongest influence on depression risk were stroke, kidney disease, hypertension, obesity, asthma, and arthritis. Information about comorbidities is usually readily available and could be useful in streamlining depression screening or targeting prevention efforts in cancer patients and survivors. A multidimensional model of the interaction between cancer and other physical comorbidities on mental health is proposed.

## 1. Introduction

Many studies show that cancer patients and survivors frequently suffer from psychological problems. Compared to the general population, individuals diagnosed with cancer are more likely to experience psychological distress or suffer common mental disorders such as depression, anxiety, adjustment, or post-traumatic stress disorder [1,2,3,4]. Depressive spectrum disorders are among the most common, with an estimated prevalence of around 16% (13% to 20%) across oncological, hematological, and palliative settings [5].

The prevalence of depression is higher among patients with a previous history of psychiatric conditions and varies as a function of the type of cancer diagnosed, the stage of disease, the treatment employed, and the time elapsed since the cancer diagnosis, among others [6,7,8,9,10,11]. Risk factors for developing depression include having lung, pancreas, head/neck or brain cancer, advanced or metastatic disease, a personal or family history of mental disorders, and being treated with certain chemotherapy agents [6]. Generally, the prevalence of depression is highest during the first year after diagnosis [8,10]. The prevalence of depression in long-term survivors (e.g., >5 years after diagnosis) has been found to be lower and similar to that of the general population without cancer history [8,9].

Another factor to consider is physical comorbidity, defined as other somatic health conditions in addition to the primary disease of interest [12]. Due to the advanced age of most cancer patients and the shared predisposing factors between cancer and other highly prevalent chronic diseases, it is common for cancer patients to suffer from one or more physical comorbidities [13]. In fact, cancer patients report more comorbidities compared to controls without cancer history [3,4]. Physical comorbidities are important in cancer care because they can limit the possibility to receive some treatments, increase their toxicity, or delay their initiation [13,14,15]. Most importantly, patients with more comorbidities have a poorer prognosis compared to those without other pre-existing conditions [14,16].

Physical comorbidities could also deteriorate the mental health of cancer patients and survivors. Depression is generally more common in adults with chronic health problems compared to people who have good physical health [17]. In cancer patients with diverse diagnoses, each additional physical comorbidity is associated with 9% higher odds of reporting high psychological distress [4].

The diagnosis of cancer is a significant life-changing event, and many cancer treatments have high toxicity or result in disfigurement, worsening the quality of life of patients. These difficult circumstances could bring about a period of high psychological vulnerability, in which co-existent physical comorbidities play a key role. Among cancer patients, both physical comorbidities and depression are significantly associated with greater healthcare utilization [4,18]. This suggests that effective prevention, screening, and management of depression in comorbid patients can help reduce healthcare utilization and cost while improving the care and quality of life of patients [18].

In the oncologic setting, depression is meant to be detected via regular screening for distress. Comprehensive clinical guidelines for the screening and management of distress in oncologic setting already exist in several countries [19]. In the U.S., the National Comprehensive Cancer Network (NCCN) Clinical Practice Guidelines in Oncology on distress management recommend that at a minimum, all patients should be screened for distress at the initial visit, at appropriate intervals, and as clinically indicated, especially with changes in disease status such as recurrence or treatment complications [20]. The NCCN Guidelines consider patients with “severe comorbid illness” to be at high risk for distress. However, it is possible that the co-existence of multimorbidity (two or more comorbidities in addition to the disease of interest) could affect distress and consequently depression risk regardless of the severity of comorbidities. For instance, conditions such as migraines, chronic back pain, and urinary incontinence have been significantly related to reporting significant psychological distress in cancer patients [4].

More detailed knowledge about to what extent, when, and for whom physical comorbidities negatively influence mental health could ultimately help streamline screening (e.g., more frequent screening in comorbid patients with specific diagnoses). This would be useful in lower resource settings where distress screening and management remains a luxury but also in higher resource setting where, despite comprehensive guidelines, a large number of patients remain unscreened [19,21].

The aim of this study was to investigate the relationship between physical comorbidities and depression in cancer survivors. Based on a previously documented relationship between physical comorbidities and psychological distress in cancer patients [4], we hypothesized that a higher physical comorbidities burden would be related to higher depression risk in cancer survivors. Given the previously documented heterogeneity in depression prevalence among survivors, we investigated how this relationship varied as a function of the time in survivorship (recent vs. long-term) and the type of cancer diagnosed. In addition, besides the total number of physical comorbidities present, we investigated to what extent only those diagnosed before the cancer predicted depression during survivorship, because they could potentially be a useful baseline assessment to guide psychological screening after diagnosis. Finally, we investigated which specific comorbidities most strongly contributed to depression.

## 2. Materials and Methods

### 2.1. Study Population

We obtained data from the National Health and Nutrition Examination Survey (NHANES), a periodic cross-sectional survey of the U.S. population [22]. The survey uses a complex, multistage probability sampling to obtain a nationally representative sample of about 5000 people each year. The NHANES samples represent the noninstitutionalized civilian population residing in the 50 states and the District of Columbia. The sample design consists of multi-year, stratified, clustered four-stage samples, with data release in 2-year cycles. The NHANES sample is drawn in four stages: (a) primary sampling units (PSUs) (counties, groups of tracts within counties, or combinations of adjacent counties), (b) segments within PSUs (census blocks or combinations of blocks), (c) dwelling units (DUs) (households) within segments, and (d) individuals within households. PSUs are sampled from all U.S. counties. 

For the current study, we combined data from the 2007 to 2018 waves (59,842 respondents), covering 12 years. The survey was approved by the National Center for Health Statistics Institutional Review Board. All study participants provided informed consent.

A flow chart of the study sample selection process is displayed in Figure 1. The inclusion criteria for the current study were having been diagnosed with cancer when 18 years old or older and having full or imputable data on the depression assessment (at least 8 out of 9 questions answered) (n = 2869). A cancer diagnosis was defined as a “yes” response to the interview question, “Have you even been told by a doctor or other health professional that you had cancer or a malignancy of any kind?”. Respondents were then excluded if they reported more than one cancer (n = 302), answered “I don’t know” to the question about the type of cancer diagnosed (n = 10), or were 80 years old or older (n = 484). This last criterion was applied because the age of individuals ≥80 is censored at 80 years for privacy concerns; however, exact age was necessary to conduct the current study. The final sample size was n = 2073 and this was further divided into recent survivors (n = 853, diagnosed in the past 5 years) and long-term survivors (n = 1220, diagnosed more than 5 years ago) following Brandenbarg et al. [9].

For a comparison of depression prevalence, one “control” sample of equal size and matched on survey wave, sex, age, and race was obtained for each sub-population from among survey participants who reported no previous cancer history. When no exact age match was available (n = 7 for recent and n = 17 for long-term survivors), age was set at ±1 year until a match was found.

### 2.2. Variables

#### 2.2.1. Sociodemographic Characteristics

Data were obtained on self-reported sex, age (continuous and grouped as 18–39, 40–59, and 60–79), race (Non-Hispanic White, Mexican American, Other Hispanic, Non-Hispanic Black, and Other), education level (up to 11th grade, high school graduate/General Educational Development (GED), some college or associate’s (AA) degree, and college graduate or above), civil status (married/living with partner, single, divorced/separated, and widowed), and health insurance status (insured vs. non-insured).

#### 2.2.2. Physical Function Limitations

The participants were categorized into three groups: “no limitations”, “limited”, and “disabled”, based on responses to several questions in the Physical Limitations module of the survey, following an established algorithm described in detail elsewhere [23]. The “disabled” group included individuals who reported needing special equipment to walk or having more than some difficulty with any of the following: walking between rooms on the same level, standing up from an armless straight chair, getting in and out of bed, eating (including holding a fork, cutting food or drinking from a glass), dressing (including tying shoes, working zippers, and doing buttons), reaching up over head, or using fingers to grasp or handle small objects. The “limited” group included individuals who only had some difficulty with the above activities or any difficulty walking up ten steps or walking for a quarter mile. The remaining individuals were allocated to the “no limitations” group.

#### 2.2.3. Cancer Site

Cancer groups were defined by anatomical site or system affected following the National Cancer Institute [24]: skin non-melanoma (SNM), melanoma, skin unknown type, breast (female), gynecological (cervix, ovary, uterus), genitourinary (prostate, bladder, kidney, testis), and digestive/gastrointestinal (colon, rectum, esophagus, gallbladder, liver, pancreas, stomach). The remaining cancer groups with very few respondents were grouped under “Other” (e.g., lung, lymphoma, leukemia, etc., see Table A1 in the Appendix A).

#### 2.2.4. Time Since Diagnosis

This was calculated as the difference between the participant’s current self-reported age and the age they reported in answer to the question “How old were you when the cancer was first diagnosed?”. This was then grouped in 1-year intervals for the sample of recent survivors and intervals of “6–10 years”, “11–20 years”, and “20+ years” for long-term survivors.

#### 2.2.5. Physical Comorbidities

We used data regarding 16 self-reported physical chronic conditions which were consistently and continuously assessed in all relevant waves of the survey: angina pectoris, arthritis, asthma, chronic bronchitis, congestive heart failure, coronary heart disease, diabetes, emphysema, gout, hypertension, liver condition, myocardial infarction, obesity, stroke, thyroid problems, and kidney disease. Participants were classified as suffering each comorbidity if they answered “Yes” to the question “Has a doctor or other health professional ever told you that you have [condition]?”. One exception was obesity, defined as a lifetime maximum self-reported body-mass index ≥30, based on questions about participants’ current height and self-reported greatest weight. The total number of comorbidities was calculated and divided into groups of “0”, “1”, “2”, and “3 or more” for descriptive purposes and groups of “0” (no comorbidity), “1” (comorbidity), and “2 or more” (multimorbidity) for analysis following [15].

#### 2.2.6. Physical Comorbidities Pre- and Post-Diagnosis

For each comorbidity, participants reported the age at which it was first diagnosed from which we determined whether it occurred before the cancer diagnosis or about at the same time or after. We then calculated the number of comorbidities participants had diagnosed before their cancer diagnosis (pre-diagnosis) and the number of comorbidities they had diagnosed at the same time or after the cancer (post-diagnosis).

#### 2.2.7. Depression

This was assessed with the Patient Health Questionnaire-9 (PHQ-9) questionnaire, which is a valid self-report instrument based on the DSM-IV criteria for diagnosis of depressive disorder [25,26]. It consists of nine items scored from 0 to 3, resulting in a total score between 0 and 27. The items assess anhedonia, depressed mood, sleep, energy, appetite, guilt and worthlessness, concentration, feeling slowed down or restless, and suicidal thoughts over the past two weeks. Previous research has established the simple scoring method of a score ≥10 as the best cut-off to detect moderate depression [27,28], hence, individuals scoring 10 or more were considered at high risk of depression. In the case of missing data (only possible on one item as per inclusion criteria), the missing value was imputed with the scale mean (in 1.8% of respondents) [29].

### 2.3. Statistical Analysis

Analyses were performed in R (v.3.6.1) [30] using the package “survey” (v.3.37) [31], and following the analytic guidelines provided by NHANES [22]. The provided sample weights (combined across the used waves) were applied in all analyses unless otherwise specified.

The two populations (recent and long-term survivors) were compared on basic demographic and health data using chi-square and *t*-tests. The prevalence of depression was estimated for each population overall and as a function of cancer site and years elapsed since diagnosis. Confidence intervals at 95% were calculated using the Korn and Graubard method as implemented in the package “survey”.

The prevalence in cancer survivors was then compared to the prevalence found in the matched control populations without cancer history using two approaches: simple conditional logistic regressions for paired case-control data (*clogit* function in the package “survival” (v. 3.1–8) [32]), which ignores the sampling weights but considers the paired nature of the data, and chi-square tests (*svychisq* function the package “survey”), which considers the sampling weights but ignores the paired nature of the data. This was done because of the lack of an analytical approach that accommodates unequal sampling weights in matched case-control data.

The relationship between comorbidities and depression (PHQ-9 ≥ 10 vs. PHQ-9 < 10) was investigated in each population using multiple logistic regression models adjusted for age, sex, race, education, civil status, health insurance status, years since diagnosis, cancer site, and physical functioning limitations. When investigating the relationship in subgroups based on diagnosis, due to the smaller sample sizes, models were adjusted only for the significant predictors in the model of the whole population. The contribution of each specific comorbidity to depression was also investigated using multiple logistic regression controlling for socio-demographic and health-related factors and including each comorbidity as a separate predictor. Analyses were conducted both including and excluding patients with SNM because this cancer is very common, rarely life threatening, and is frequently considered as a different entity.

## 3. Results

Demographic data and comparisons between short and long-term survivors are shown in Table 1. Compared to short-term survivors, long-term survivors were more likely to be female, older, and from an ethnic minority classified as “other”. The prevalence of SNM cancer was higher among recent survivors, whereas the prevalence of gynecological cancers was higher among long-term survivors (Table 1). The two groups did not differ on the number of total comorbidities, however, recent survivors reported a significantly larger number of comorbidities diagnosed before the cancer, whereas long-term survivors reported a larger number of comorbidities diagnosed after the cancer.

### 3.1. Recent Survivors

The prevalence of depression (PHQ-9 ≥ 10) was 10.3% (95% CI 7.2–13.5) in the population of recent survivors (excluding SNM, see Table 2). Except for diagnoses of SNM, “other”, and “skin unknown type” cancers, the prevalence of depression was consistently higher in recent survivors compared to controls, albeit significantly so only during the first year after diagnosis and for gynecological cancers (Table 2).

The most prevalent comorbidities in this population were obesity, hypertension, arthritis, asthma, diabetes, thyroid problems, and chronic bronchitis (see Table A2 in the Appendix A). The prevalence of depression was higher among survivors who reported more comorbidities: the prevalence among those reporting no comorbidities was 1.9%, compared to 3.3% among those reporting one, 5.7% two, and 13.6% three or more comorbidities.

Total number of comorbidities. In multiple regression controlling for sociodemographic and health-related factors, the total number of comorbidities was significantly related to depression: each additional comorbidity increased the odds of suffering depression according to the PHQ-9 by about 50% (OR = 1.54, 95% CI 1.29–1.84, see Table 3). Suffering one comorbidity vs. none did not significantly increase the risk of depression, OR = 2.41, 95% CI 0.54–10.72, but suffering multimorbidity (two or more comorbidities) did with OR = 8.12, 95% CI 2.23–29.60 in the whole sample and OR = 6.75, 95% CI 1.80–25.25 excluding survivors of SNM. 

The relationship between comorbidities and depression was strongest among recent survivors of breast cancer, followed by genitourinary and gynecological cancers.

Besides comorbidities, the other significant predictors of depression were age and physical function limitations. Younger patients were more likely to report depression with OR = 3.74, 95% CI 1.70–8.21 for the 40–59 vs. 60+ group, and OR = 10.37, 95% CI 3.17–33.90 for the 20–39 vs. 60+ group. Survivors who were disabled were also more likely to report depression compared to survivors without physical limitations, OR = 3.60, 95% CI 1.41–9.19.

Comorbidities diagnosed before the cancer. The results were similar when only the comorbidities diagnosed before the cancer were considered, both in the total sample and without considering survivors of SNM (Table 3). Suffering one comorbidity vs. none before the cancer was diagnosed did not significantly increase the risk of depression, OR = 0.57, 95% CI 0.15–2.20, but suffering multimorbidity did with OR = 3.48, 95% CI 1.26–9.55 in the whole sample, and OR = 7.11, 95% CI 1.69–29.97 excluding patients with SNM.

Comorbidities diagnosed after/at the same time as the cancer. The number of comorbidities diagnosed after the cancer was not significantly related to depression, OR = 1.21, 95% CI 0.67–1.87.

Effects of specific comorbidities. The specific comorbidities that were most strongly and significantly related to depression were stroke, kidney disease, hypertension, obesity, asthma, and arthritis (Figure 2).

### 3.2. Long-Term Survivors

The prevalence of depression (PHQ-9 ≥ 10) was 9.7% (95% CI 7.7–11.7) in the population of long-term survivors (excluding SNM, see Table 2). The prevalence of depression was higher in long-term survivors compared to controls, in particular after 10 or more years of survivorship and for survivors of gynecological and skin unknown type cancers (Table 2).

The most prevalent comorbidities in this population were similar to those found in short-term survivors: obesity, arthritis, hypertension, thyroid problems, asthma, diabetes, and chronic bronchitis (see Table A2). The prevalence of depression was higher among long-term survivors who reported more comorbidities: the prevalence among those reporting no comorbidities was 3.0%, compared to 7.8% among those reporting one, 8.6% two, and 10.6% three or more comorbidities.

However, in multiple regression controlling for sociodemographic and health-related factors, the number of comorbidities was not significantly related to depression, OR = 1.09, 95% CI 0.97–1.23. This was the case also when only the number of comorbidities diagnosed before or after the cancer was considered, in the total population, and excluding survivors of SNM (all 95% CIs containing 1).

Instead, the significant predictors of depression in this population were age, education, and physical function limitations. Younger patients were more likely to report depression with OR = 3.92, 95% CI 2.05–7.49 for the 40–59 vs. 60+ group, and OR = 11.07, 95% CI 3.73–32.84 for the 20–39 vs. 60+ group. Compared to those who were college graduates or above, survivors who had finished high school or less were more likely to report depression, OR = 3.40, 95% CI 1.64–7.03. Survivors who were disabled were also more likely to report depression compared to survivors without physical limitations, OR = 5.57, 95% CI 3.03–10.22.

## 4. Discussion

Comorbidities were strongly related to depression as measured by the PHQ-9 questionnaire in recent but not in long-term survivors. This was the case both when considering the total number of comorbidities present at the time of the survey and only the comorbidities diagnosed before the cancer. In particular, recent survivors who suffered multimorbidity (two or more physical conditions) before their cancer was diagnosed had 3 to 4 times the odds of reporting significant depressive symptoms up to 5 years after the cancer diagnosis.

This suggests that the number of comorbidities present at the time of diagnosis could be a very important predictor of depression risk up to five years after cancer diagnosis, independent of many demographics and health-related factors such as sex, age, race, or physical functioning limitations. This possibility should be explored in prospective studies and based on confirmed depression diagnoses, and if the current findings are confirmed, the presence of additional comorbidities, in particular multimorbidity, could be used to target prevention efforts or streamline screening for depression during treatment and follow up. Information about comorbidities is usually readily available in medical records and always should be considered during cancer treatment, so no changes would be required to the screening instruments already in use. The prevention and early detection of depression in the cancer setting is of highest importance because patients who develop depression after their cancer diagnosis have shorter survival [33].

Comorbidities were significantly related to depression in SNM, breast, gynecological, and genitourinary cancer survivors. The relationship was strongest in breast cancer survivors, such that each additional comorbidity more than doubled the odds of reporting significant depressive symptoms. These results support the findings of a systematic review which identified comorbidity as a significant factor for distress in breast cancer survivors [34]. In contrast, no relationship was found for survivors of digestive/gastrointestinal cancers, melanoma, skin unknown type cancers, and other cancers.

The comorbidities with the strongest independent effects on depression risk in recent survivors were stroke, kidney disease, hypertension, obesity, asthma, and arthritis, all of which have been previously linked to depression [35,36,37,38,39]. Having in mind that depression risk in this study was driven by multimorbidity, the different comorbidities may interact in important ways. For instance, the clustering of obesity, hypertension, and diabetes into the metabolic syndrome has been associated with depression [37]. Future studies should explore the most detrimental combinations of diseases when it comes to mental health outcomes in cancer survivors.

There are multiple biological, physical, social, and psychological mechanisms through which physical comorbidities could increase the risk of developing depression after a cancer diagnosis (see Figure 3). The comorbidities with the strongest independent effects on depression risk in the current study (Figure 2) included multiple conditions associated with chronic inflammation, suggesting it could be one of the principal mechanisms. A chronic state of inflammation has been implicated in the pathogenesis of depression [40,41,42], multiple types of cancer including breast, colorectal, lung, ovarian, and prostate cancer [43], and the chronic conditions that most frequently co-exist in cancer patients (i.e., cardiovascular conditions, obesity, metabolic, and autoimmune disorders) [44,45,46].

Another related biological mechanism is increased allostatic load: the cumulative biological burden on the body’s systems due to repeated adaptation to stressors over time, reflected in cardiovascular, metabolic, and immunological system parameters [47,48]. Finally, some medications for cancer are known to increase depression risk [6,40] and the medications used to treat the co-existing conditions could interact in unknown ways to increase this risk.

Physical mechanisms include more physical limitations, higher symptom burden, and lower quality of life among patients with more comorbidities [49,50]. In the current study, the relationship between comorbidities was significant even when controlling for physical function limitations, which is one potential mechanism through which comorbidities could influence mental health. In other words, knowing the patient’s physical limitations status (i.e., whether they are disabled to a certain extent) would be useful but not optimal to predict depression risk and it would be worth exploring the other potential mechanisms.

Finally, social and psychological mechanisms include the experience of different social and personal limitations due to multi-morbidity [40], lower self-esteem resulting from disease labeling or inability to fulfil social or family functions [40,51,52], or lower hopes for the future due to worse cancer prognosis [14]. In summary, the usually readily available and simple information about the number of comorbidities could capture and summarize the multidimensional underlying biological, social, and psychological risk.

Comorbidity was not significantly related to depression in long-term survivors, even when physical functioning—a potential underlying mechanism—was not included in the model. This suggests that information about comorbidity may be of little utility for screening of long-term survivors, at least those without second cancers and who are well enough to participate in an extensive survey. It appears that comorbidity may have an effect on mental health only during the “window of psychological vulnerability” produced after the cancer diagnosis when the physical, psychological, and social impact of the cancer diagnosis and treatment is highest.

The mental health of cancer survivors during long-term survivorship is much less researched compared to the first few years after diagnosis [9]. On one hand, research shows that once beyond the symbolic 5-year mark, when most cancer follow-up schedules are discontinued, individuals begin to identify as “survivors” or “ex-patients” as opposed to “patients” or “victims” [52]. This transformation has been related to the disappearance of symptoms and disability and the reception of assuring test results after treatment success [52]. Perhaps this change in disease-related identify and the associated increase in well-being could buffer the negative effect of comorbidities in long-term survivors. On the other hand, cancer patients who suffer multimorbidity [14] and depression [33] after their cancer diagnosis have higher mortality rates. Thus, individuals who are psychologically the most vulnerable to the clustering of different comorbidities may be less likely to make it into long-term survivorship.

Contrary to the results of a recent review [9], we found that the prevalence of depression was higher in long-term survivors compared to controls matched on several demographic characteristics. Interestingly, this was the case for survivors beyond 10 years from diagnosis, a period that is much less frequently researched. Cancer patients can suffer diverse physical and psychological sequelae and symptoms for longer than 10 years after treatment [53]. This could contribute to the higher rates of depression found in this population, which was nevertheless not related to the presence of comorbidities.

Depression was also significantly more prevalent among survivors of gynecological cancers regardless of the time elapsed after diagnosis. Gynecological cancer survivors were the youngest in the current study (mean age of 47 and 55 in short and long-term survivors, respectively). Survivors of gynecological cancers experience diverse persistent sequelae such as sexual problems, premature menopause, pain, and fatigue, which can have implications for quality and life and ultimately mental health [54].

Limitations of this study include the use of a self-report instrument instead of full clinical interview as a measure of depression. No information was available regarding the stage of disease at the time of diagnosis, or the treatments received, and individuals 80 years old or older had to be excluded. We did not control for lifestyle factors such as alcohol use, nutrition, exercise, or sleep that could also influence depression risk [55].

There was only a small number of survivors with certain cancer diagnoses which prevented us from investigating the role of comorbidities in patients diagnosed with these tumors (e.g., head and neck, pancreatic, lung). Some of these cancers have very low survival rates and hence the prevalence of survivors in the population would be also low, especially in the long-term group. In addition, whereas the total sample size of survivors was relatively large, subgroup analyses of the different types of cancer were based on a much smaller number of respondents (between 74 and 145). Hence, a small or medium effect of comorbidities on depressive symptoms may exist also in the subgroups where it was not significant; however, most subgroup analyses conducted were only powered to detect large effect sizes (see Sensitivity analyses on OSF: https://osf.io/uqcy4/, doi 10.17605/OSF.IO/UQCY4, accessed on 5 July 2021).

The assessment of comorbidities and the cancer diagnosis could be affected by recall bias. Previous studies based on NHANES and other cohorts examined the validity of self-reported diagnoses of diverse chronic conditions against medical records or cancer registry data [56,57]. These studies indicate that accuracy, although generally acceptable, varies strongly by condition. In the NHANES waves used in the current study, to assess when each condition was diagnosed, respondents had to indicate how old they were when it was first diagnosed. We believe that this general question, together with the fact that a cancer diagnosis is usually a very impactful and thus memorable event, provides an effective benchmarking technique to determine which comorbidities were present before and which after the cancer diagnosis. However, the validity of this self-reported method should be examined empirically in future studies and ideally multiple sources should be used to achieve the highest possible accuracy [56,57].

Cancer patients with previous history of a psychiatric disorder, current depression, or a substance use disorder are considered at higher risk for psychological distress following a cancer diagnosis [20]. However, no information regarding previous history of the psychiatric conditions of psychological distress was available. In future studies, it would be important to address the joint effects of physical and psychiatric comorbidities on depression risk, as these could interact in important ways. Finally, there could be further physical comorbidities that contribute to depression but were not assessed in the survey.

## 5. Conclusions

In conclusion, multimorbidity before and after the cancer diagnosis was significantly related to depression only in recent cancer survivors (up to 5 years after diagnosis). The relationship was strongest for breast cancer followed by genitourinary, and gynecological cancer survivors. Future research should explore the utility of information about comorbidities in the triage of screening and full clinical evaluation for depression in cancer survivors.

## Figures and Tables

**Figure 1 cancers-13-03368-f001:**
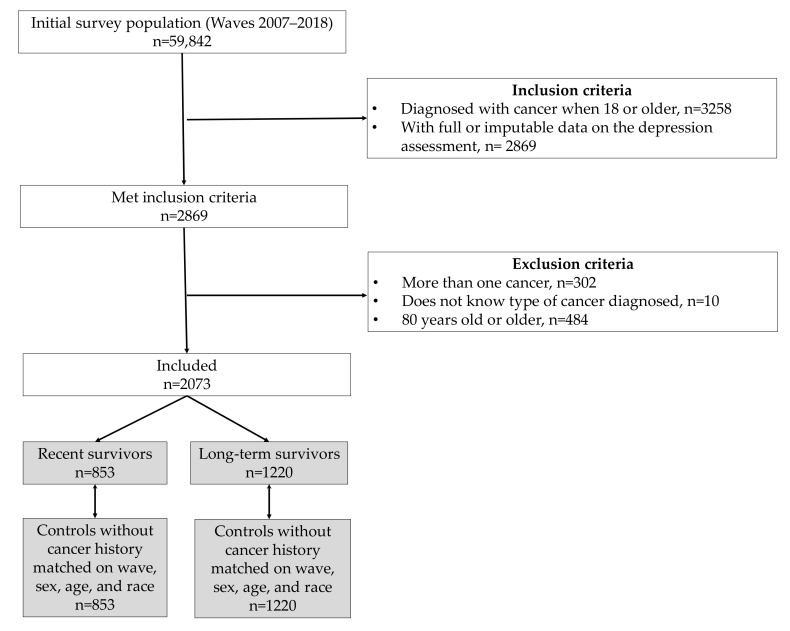
Flow chart describing the study sample selection process.

**Figure 2 cancers-13-03368-f002:**
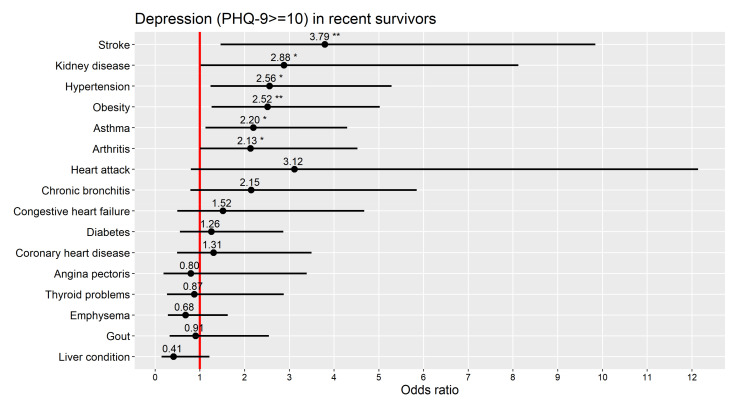
Odds ratios (black dots) and their 95% confidence intervals (black lines crossing the dots) for individual comorbidities from multiple regression analysis in recent survivors with outcome depression (PHQ-9 ≥ 10) and adjusted for age, sex, race, and years since diagnosis. Red line: OR = 1 (no effect). Note: * *p* ≤ 0.05. ** *p* ≤ 0.01. * and ** also denote confidence intervals excluding 1.

**Figure 3 cancers-13-03368-f003:**
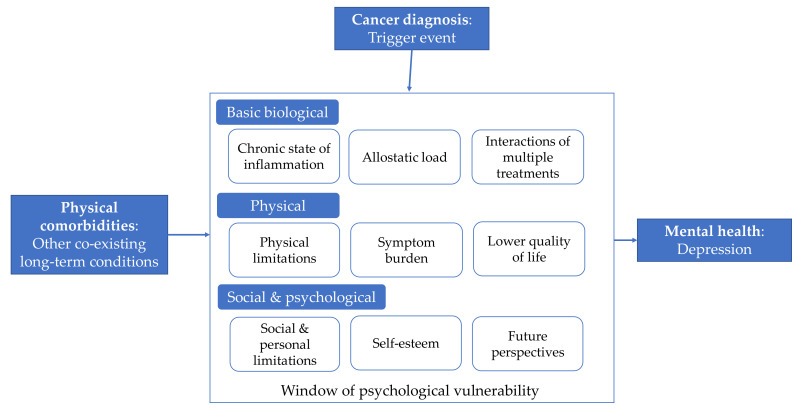
Theoretical model of potential mechanisms through which physical comorbidities could increase the risk of mental health problems such as depression following a cancer diagnosis.

**Table 1 cancers-13-03368-t001:** Basic demographic and clinical characteristics of recent and long-term cancer survivors.

Variable	Category	Recent Survivors n = 853	Long-Term Survivors n = 1220	*p*
Sex	Male	49.4	40.9	0.007
	Female	50.6	59.1	
Age	M ± SD	58.6 (13.1)	61.2 (11.9)	0.001
	18–39	10.4	6.9	
	40–59	37.2	31.4	
	60–79	52.4	61.7	
Race	Non-Hispanic White	85.3	83.9	0.006
	Non-Hispanic Black	6.7	5.3	
	Other Hispanic	2.8	2.8	
	Mexican American	2.7	2.9	
	Other	2.5	5.1	
Education	Less than 9th grade	2.8	3.2	0.707
	9–11th grade	7.8	6.6	
	High school graduate/GED	20.8	19.5	
	Some college or AA degree	33.1	32.5	
	College graduate or above	35.5	38.2	
Civil status	Married or living with partner	70.1	68.2	0.052
	Divorced or separated	14.8	17.3	
	Single, never married	8.2	5.2	
	Widowed	6.9	9.3	
Insurance status	Insured	95.4	92.8	0.095
	Uninsured	4.6	7.2	
Cancer site	Skin non-melanoma	27.7	19.7	<0.001
	Melanoma	6.8	7.7	
	Skin unknown type	8.6	8.5	
	Breast	13.8	15.2	
	Gynecological	7.9	18.6	
	Genitourinary	13.8	12.7	
	Gastrointestinal	6.2	4.4	
	Other	15.2	13.1	
Years since diagnosis	1 year	35.2	-	-
	2 years	18.4	-	
	3 years	17.1	-	
	4 years	14.3	-	
	5 years	15.0	-	
	6–10 years	-	40.6	
	11–20 years	-	34.9	
	20+ years	-	24.5	
Comorbidities total	M ± SD	2.61 (2.06)	2.67 (2.01)	0.704
	None	15.6	10.2	
	One	17.8	22.7	
	Two	19.9	20.1	
	Three or more	46.7	47.0	
Comorbidities pre-diagnosis	M ± SD	2.45 (1.83)	1.21(1.32)	<0.001
	None	10.1	35.8	
	One	24.5	32.6	
	Two	23.9	16.5	
	Three or more	41.5	15.0	
Comorbidities post-diagnosis	M ± SD	0.55 (0.78)	1.84 (1.44)	<0.001
	None	58.2	14.5	
	One	31.9	33.4	
	Two or more	9.9	52.1	
Physical function limitations	No limitations	64.6	62.2	0.431
	Limited	20.1	23.2	
	Disabled	15.3	14.6	
Depression (PHQ-9 ≥ 10)	Depressed	8.4	8.8	0.754
	Not depressed	91.6	91.2	

Note: Values are percentages for categorical variables and mean and standard deviation, M(SD), for age and number of comorbidities. *p*-values are from chi-square tests comparing recent and long-term survivors for categorical variables and from *t*-tests for age and the number of comorbidities.

**Table 2 cancers-13-03368-t002:** Prevalence (percentages and 95% confidence intervals, CIs) of depression (PHQ-9 ≥ 10) according to cancer site and years since diagnosis in each subpopulation of survivors and their controls.

		Recent Survivors					Long-Term Survivors				
		Cases			Matched Controls		Cases			Matched Controls	
		Total N	Observed N with PHQ-9 ≥ 10	Prevalence (weighted % and 95% CI)	Observed N with PHQ-9 ≥ 10	Prevalence (weighted % and 95% CI)	N	Observed N with PHQ-9 ≥ 10	Prevalence (weighted % and 95% CI)	Observed N with PHQ-9 ≥ 10	Prevalence (weighted % and 95% CI)
Cancer site	All	853	89	8.4 (6.1, 11.2)	70	7.5 (5.5, 9.9)	1220	139	8.8 (6.9, 11.0) *^	89	5.6 (3.9, 7.6)
	All except skin non-mel.	708	83	10.3 (7.4, 13.9)	61	7.7 (5.5, 10.5)	1059	131	9.7 (7.7, 11.9) *^	78	6.0 (4.2, 8.3)
	Skin non-melanoma	145	6	3.2 (0.7, 9.1)	9	6.7 (2.1, 15.3)	161	8	5.5 (2.2, 11.0)	11	3.3 (1.4, 6.5)
	Melanoma	44	2	6.6 (0.1, 37.2)	3	3.8 (0.1, 18.1)	70	7	8.9 (2.2, 22.4)	10	11.1 (2.1, 30.6)
	Skin unknown type	60	6	5.9 (0.5, 21.3)	4	6.9 (0.4, 27.6)	88	12	12.6 (4.8, 25.4) ^	5	2.1 (0.6, 5.5)
	Breast	128	22	16.6 (7.8, 29.3)	16	11.7 (4.8, 22.7)	202	23	8.6 (3.5, 17.0)	12	3.6 (1.4, 7.5)
	Gynecological	74	18	19.1 (7.6, 36.6) *	6	7.6 (1.3, 23.3)	229	44	14.0 (8.9, 20.7) *^	21	8.1 (4.3, 13.6)
	Genitourinary	189	11	7.0 (2.6, 14.6)	11	6.1 (2.6, 12.0)	216	14	4.1 (1.8, 8.1)	13	4.2 (1.9, 8.0)
	Digestive/gastrointest.	79	13	11.9 (4.7, 23.6)	6	7.9 (3.0, 16.2)	83	14	12.7 (4.8, 25.7)	8	10.1 (1.8, 28.6)
	Other	134	11	6.7 (2.4, 14.6)	15	7.5 (3.2, 14.5)	171	17	7.5 (3.9, 12.9)	9	5.0 (1.6, 11.5)
Time since diagnosis (all except skin non-mel.)	1 year	238	29	12.4 (7.1, 19.6)^	17	4.4 (2.3, 7.6)			-		
	2 years	139	17	9.1 (4.0, 17.2)	12	10.3 (3.1, 23.4)			-		
	3 years	117	15	10.9 (4.0, 22.8)	12	10.0 (4.4, 18.8)			-		
	4 years	94	13	8.8 (3.9, 16.6)	7	6.9 (2.3, 15.3)			-		
	5 years	120	9	8.1 (4.5, 13.2)	13	9.5 (4.6, 16.9)			-		
	6–10 years			-			430	40	7.9 (5.9, 10.2)	31	6.0 (3.4, 9.6)
	11–20 years			-			383	59	11.4 (7.8, 15.9) *^	30	5.6 (3.0, 9.3)
	>20 years			-			246	32	10.1 (5.7, 16.1) *	17	6.6 (2.8, 12.9)

Note: * Significantly different from controls (*p* value ≤ 0.05 from model Likelihood ratio tests) based on simple conditional logistic regression for paired case-control data. ^ Significantly different from controls (*p* value ≤ 0.05) based on chi-square test in the “survey” package. Note that observed N/total N does not equal the estimated prevalence, which is derived using the sampling weights.

**Table 3 cancers-13-03368-t003:** Effect of comorbidities on depression (PHQ-9 ≥ 10) overall and according to cancer site in recent cancer survivors (n = 853).

Table		Comorbidities Total	Comorbidities Pre-Diagnosis
	N	OR (95% CI)	OR (95% CI)
All *	853	**1.54 (1.29, 1.84)**	**1.51 (1.22, 1.87)**
All except skin non-melanoma *	708	**1.57 (1.30, 1.89)**	**1.63 (1.30, 2.03)**
Skin non-melanoma **	145	1.52 (1.00, 2.30)	**1.45 (1.10, 1.91)**
Skin (melanoma and skin unknown type) **	104	1.39 (0.87, 2.20)	1.39 (0.87, 2.21)
Breast **	128	**2.49 (1.63, 3.81)**	**2.32 (1.60, 3.36)**
Gynecological **	74	**1.53 (1.12, 2.09)**	1.40 (0.96, 2.05)
Genitourinary **	189	**1.61 (1.20, 2.16)**	**1.77 (1.16, 2.70)**
Digestive/gastrointestinal **	79	0.95 (0.65, 1. 39)	1.02 (0.63, 1.64)
Skin **	104	1.39 (0.87, 2.20)	1.39 (0.87, 2.21)
Other **	134	1.32 (0.84, 2.06)	1.14 (0.78, 1.66)

Note: * From a multiple regression model controlling for age, sex, race, education, civil status, insurance status, years since diagnosis, cancer site, and physical functioning limitations. ** From a multiple regression model controlling for age and physical functioning limitations. Melanoma and skin unknown type were combined for this analysis due to their small sample sizes but similarity in depression prevalence.

## Data Availability

The data used for this research are publicly available on the NHANES website: https://www.cdc.gov/nchs/nhanes/index.htm, accessed on 5 July 2021. The dataset and analysis for this specific study are available on the Open Science Framework (OSF): doi:10.17605/OSF.IO/UQCY4, https://osf.io/uqcy4/, accessed on 5 July 2021.

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
