# Peer review of "Physical Comorbidities and Depression in Recent and Long-Term Adult Cancer Survivors: NHANES 2007–2018"

_cancers, 2021, doi:10.3390/cancers13133368_

Round 1

Reviewer 1 Report

Thank you for opportunity to review the manuscript titled "Physical comorbidities and depression in recent and long-term adult cancer survivors: NHANES 2007-2018.". The work is well written and clarifies the results clearly. The great adventage is large sample. Tests were well applied. The conclusions are correct.
Few minor points:
1) The studied population is very specific. The is great number of skin cancer survivors (about 40% of studied group), but no lung cancer, laryngeal cancer an many others. The explanation is needed.
2) Has the alcohol use was studied? That might affect the results.
3) Have gender effects on outcomes been excluded? The relationship was strongest in breast cancer survivors.
4) Stroke effected the depression risk in recent survivors. Has the impact of disability on outcomes been  investigated?

Author Response

Reviewer 1

Thank you for opportunity to review the manuscript titled "Physical comorbidities and depression in recent and long-term adult cancer survivors: NHANES 2007-2018.". The work is well written and clarifies the results clearly. The great advantage is large sample. Tests were well applied. The conclusions are correct.

A: Thank you for the positive and constructive feedback on the manuscript.

Few minor points:

1) The studied population is very specific. The is great number of skin cancer survivors (about 40% of studied group), but no lung cancer, laryngeal cancer and many others. The explanation is needed.

A: Thank you for raising this point. We have included an explanation in the limitations section of the manuscript. Some cancers such as non-melanoma skin cancer have very high survival rates, so the prevalence of survivors of such cancers is higher in the population. As a consequence, they are more likely to end up as respondents of a national survey.

“There was only a small number of survivors with certain cancer diagnoses which prevented us from investigating the role of comorbidities in patients diagnosed with these tumors (e.g., head and neck, pancreatic, lung). Some of these cancers have very low survival rates and hence the prevalence of survivors in the population would be also low, especially in the long-term group.”

2) Has the alcohol use was studied? That might affect the results.

A: We have not studied the effect of alcohol use but we agree that it would be important to do so in future studies, both in terms of regular consumption or as a substance abuse problem:

“We did not control for lifestyle factors such as alcohol use, nutrition, exercise, or sleep that could also influence depression risk [55].”

“Cancer patients with previous history of a psychiatric disorder, current depression, or a substance use disorder are considered at higher risk for psychological distress following a cancer diagnosis [20]. However, no information regarding previous history of psychiatric conditions of psychological distress was available. In future studies, it would be important to address the joint effects of physical and psychiatric comorbidities on depression risk, as these could interact in important ways.”

3) Have gender effects on outcomes been excluded? The relationship was strongest in breast cancer survivors.

A: We have adjusted for sex in the adjusted multiple regression models used to study the effect of comorbidities on depression (see Table 3), however, it was not a significant predictor neither in the population of short-term, nor in the population of long-term survivors. The breast and gynecological subgroups include only women, whereas the genitourinary group is more male dominant (due to the high prevalence of prostate cancer survivors in this group). Effects of comorbidities on depression were found in all these subgroups, but the effect was strongest in the breast cancer subgroup. Hopefully, future research can clarify how dimensions related to sex, cancer treatment sequalae or other social factors contribute to the effect of comorbidities on depression risk.

4) Stroke effected the depression risk in recent survivors. Has the impact of disability on outcomes been investigated?

A: Yes, using the variable called “physical functioning limitations”, which divides survey respondents into three groups: “disabled”, “limited”, and “no limitations” (see page 4, 2.2.2. Physical function limitations). This variable has been adjusted for in all regression models in Table 3. Also, it had a strong effect on depression symptoms in both short-term and long-term survivors, such that survivors who were disabled had a much higher depression risk compared to the group without limitations (described in the respective Results sections for short-term and long-term survivors). The effect of comorbidities was independent of the effect of physical functioning limitations.

As for the effect of stroke in particular, it is slightly diminished when the effect of physical functioning limitations is added in the model, however it remains significant with OR=2.72, 95% CI 1.04-7.08.

Reviewer 2 Report

Title: Physical comorbidities and depression in recent and long-term adult cancer survivors: NHANES 2007-2018

It is an interesting topic and the role of comorbidity is important to consider. The paper is well written and presented.

Comments:

-There is no information available on mental disorders or distress level of patients before the cancer diagnosis. There is literature about that, so please write a paragraph in the discussion about the role of physical comorbidities, mental comorbidities and the effect of both, to put these findings into perspective.

-Please list some of your hypotheses and argue why you state them.

-what is the role of recall bias? How certain is it that patients remember when there comorbidities occurred. Please elaborate on this phenomenon and limitation in the discussion.

-The sample size calculation is lacking. Is this number sufficient to adjust for all (9) confounders? Especially some sub groups are very small.

-information of the sample selection and representativeness is lacking and should be added in the methods section. In addition it should be elaborated on in the discussion and if there is a power problem for some groups it should be listed.

-please list N of depressive cases in Table 2 and not only percentages.

-paragraph 3.1: why do the authors so explicitly state the depression prevalence excluding SNM survivors. If this is so important, please mention the rationale.

-page 9 sentence line 301-302 is unclear. Better specify throughout the text if the authors studies comorbidity before or after the cancer diagnosis or both.

-Figure 2: it is unclear whether 1 falls within the confidence interval for kidney disease and arthritis, please make this more clear.

-please provide a more thorough explanation with relevant literature on why the effects were only observed for short-term and not for long-term survivors.

-was there any clinical data about treatment available?

Author Response

Reviewer 2

Comments and Suggestions for Authors

Title: Physical comorbidities and depression in recent and long-term adult cancer survivors: NHANES 2007-2018

It is an interesting topic and the role of comorbidity is important to consider. The paper is well written and presented.

A: Thank you for the positive and constructive feedback on the manuscript.

Comments:

-There is no information available on mental disorders or distress level of patients before the cancer diagnosis. There is literature about that, so please write a paragraph in the discussion about the role of physical comorbidities, mental comorbidities and the effect of both, to put these findings into perspective.

A: We have addressed this in the discussion: “Cancer patients with previous history of a psychiatric disorder, current depression, or a substance use disorder are considered at higher risk for psychological distress following a cancer diagnosis [20]. However, no information regarding previous history of psychiatric conditions of psychological distress was available. In future studies, it would be important to address the joint effects of physical and psychiatric comorbidities on depression risk, as these could interact in important ways.”

-Please list some of your hypotheses and argue why you state them.

A: We only had one general hypothesis, which we have now included and justified in the final paragraph in the introduction:

“Based on a previously documented relationship between physical comorbidities and psychological distress in cancer patients [4], we hypothesized that a higher physical comorbidities burden would be related to higher depression risk in cancer survivors.”

We had no specific hypotheses regarding this relationship in the sub-groups of survivors, hence these analyses were exploratory.

-what is the role of recall bias? How certain is it that patients remember when there comorbidities occurred? Please elaborate on this phenomenon and limitation in the discussion.

A: Indeed, recall bias with regards to the diagnoses and the time when they occurred is a possible limitation. We have included discussion about this in the limitations section:

“The assessment of comorbidities and the cancer diagnosis could be affected by recall bias. Previous studies based on NHANES and other cohorts examined the validity of self-reported diagnoses of diverse chronic conditions against medical records or cancer registry data [56,57]. These studies indicate that accuracy, although generally acceptable, varies strongly by condition. In the NHANES waves used in the current study, to assess when each condition was diagnosed, respondents had to indicate how old they were when it was first diagnosed. We believe that this general question, together with the fact that a cancer diagnosis is usually a very impactful and thus memorable event, provides an effective benchmarking technique to determine which comorbidities were present be-fore and which after the cancer diagnosis. However, the validity of this self-reported method should be examined empirically in future studies and ideally multiple sources should be used to achieve the highest possible accuracy [56,57].  “

-The sample size calculation is lacking. Is this number sufficient to adjust for all (9) confounders? Especially some sub groups are very small.

A: In Table 3, in the analyses of the all recent survivors (n=853) and all recent survivors except for those of skin non-melanoma (n=708) we adjusted for all 9 variables. However, in the other analyses of sub-groups, due to the sample size concerns you mentioned, we only adjusted for age and physical functioning limitations (the only other significant predictors besides comorbidities in the analysis of all recent survivors). So, in the analyses for the specific cancer locations we only included 3 predictors in the model. This is acknowledged in the method section and in the legend of Table 3:

“When investigating the relationship in subgroups based on diagnosis, due to the smaller sample sizes, models were adjusted only for the significant predictors in the model of the whole population.”

Because the sample size was predetermined by the data constraints (i.e., how many respondents met the inclusion criteria from all survey participants), to address the concerns about its size we have conducted a sensitivity analysis in G*power. This analysis calculates the minimal population effect size that could be reasonably detected with the current number of respondents and analysis.

We have done this analysis for each subpopulation (recent and long-term survivors). The results show that with the current analysis and with an alpha level of 0.05 and statistical power of 0.95, in the population of recent cancer survivors, the analysis can detect an OR≥1.6 in recent survivors and OR≥1.47 in long-term cancer survivors.

We conducted the same analysis for the smallest subgroup in which we have examined the effect of comorbidities (n=74, recent survivors of gynecological cancers). In this case, as is to be expected, a much larger underlying effect size in the population is needed (OR≥3.66).

These analyses have been added to the OSF link of the study: DOI 10.17605/OSF.IO/UQCY4. We also discuss sample size issues in the Discussion as suggested.

“In addition, whereas the total sample size of survivors was relatively large, subgroup analyses of the different types of cancer were based on a much smaller number of respondents (between 74 and 145). Hence, a small or medium effect of comorbidities on de-pressive symptoms may exist also in the subgroups where it was not significant; however, most subgroup analyses conducted were only powered to detect large effect sizes (see Sensitivity analyses on OSF: doi 10.17605/OSF.IO/UQCY4).”

-information of the sample selection and representativeness is lacking and should be added in the methods section. In addition it should be elaborated on in the discussion and if there is a power problem for some groups it should be listed.

A: This information has been added as suggested on page 3: “The NHANES samples represent the noninstitutionalized civilian population residing in the 50 states and the District of Columbia. The sample design consists of multi-year, stratified, clustered four-stage samples, with data release in 2-year cycles. The NHANES sample is drawn in four stages: (a) primary sampling units (PSUs) (counties, groups of tracts within counties, or combinations of adjacent counties), (b) segments within PSUs (census blocks or combinations of blocks), (c) dwelling units (DUs) (households) within segments, and (d) individuals within households. PSUs are sampled from all U.S. counties.”

Power issues for some subgroup analyses have also been mentioned: “In addition, whereas the total sample size of survivors was relatively large, subgroup analyses of the different types of cancer were based on a much smaller number of respondents (between 74 and 145). Hence, a small or medium effect of comorbidities on de-pressive symptoms may exist also in the subgroups where it was not significant; however, most subgroup analyses conducted were only powered to detect large effect sizes (see Sensitivity analyses on OSF: doi 10.17605/OSF.IO/UQCY4).”

-please list N of depressive cases in Table 2 and not only percentages.

We have now listed those in Table 2. We also added a note to the Table legend clarifying to readers that the percentages derived from the observed number of depressive cases / total observed population does not match the prevalence estimates provided because these are derived after applying the sample weights to derive population estimates (in the R package survey).

-paragraph 3.1: why do the authors so explicitly state the depression prevalence excluding SNM survivors. If this is so important, please mention the rationale.

A: This is now explained on p. 6: “Analyses were conducted both including and excluding patients with SNM because this cancer is very common, rarely life threatening, and is frequently considered as a different entity.”

Skin non-melanoma cancer is usually excluded from most epidemiological studies on cancer and is frequently treated as a different entity (e.g., it is excluded from most official cancer statistics and many cancer registries do not even keep track of it). SNM cancer is usually diagnosed and treated easily in a doctor’s office and has a much better prognosis compared to other cancers (it is rarely considered life-threatening). However, we decided to include it in the current study because 1) patients still get a cancer diagnosis, 2) patients could also suffer some disfigurement (i.e., important scarring on the face) due to treatment, and 3) they are at an increased risk for other cancers, all of which could have some consequence for mental health and quality of life.

-page 9 sentence line 301-302 is unclear. Better specify throughout the text if the authors studies comorbidity before or after the cancer diagnosis or both.

A: Thank you, we have clarified this issue using subsection headings.

-Figure 2: it is unclear whether 1 falls within the confidence interval for kidney disease and arthritis, please make this more clear.

A: We have clarified in the figure legend that the presence of a star (* or **) in the figure denotes confidence intervals excluding 1 (the case for both kidney disease and arthritis).

-please provide a more thorough explanation with relevant literature on why the effects were only observed for short-term and not for long-term survivors.

A: This has been added to the Discussion as suggested:

“Comorbidity was not significantly related to depression in long-term survivors, even when physical functioning – a potential underlying mechanism – was not included in the model. This suggests that information about comorbidity may be of little utility for screening of long-term survivors, at least those without second cancers and who are well enough to participate in an extensive survey. It appears that comorbidity may have an effect on mental health only during the “window of psychological vulnerability” produced after the cancer diagnosis when the physical, psychological, and social impact of the cancer diagnosis and treatment is highest.

The mental health of cancer survivors during long-term survivorship is much less researched compared to the first few years after diagnosis [9]. On one hand, research shows that once beyond the symbolic 5-year mark, when most cancer follow-up schedules are discontinued, individuals begin to identify as “survivors” or “ex-patients” as opposed to “patients” or “victims” [52]. This transformation has been related to the disappearance of symptoms and disability and the reception of assuring test results after treatment success [52]. Perhaps this change in disease-related identify and the associated increase in well-being could buffer the negative effect of comorbidities in long-term survivors. On the other hand, cancer patients who suffer multimorbidity [14] and depression [33] after their cancer diagnosis have higher mortality rates. Thus, individuals who are psychologically most vulnerable to the clustering of different comorbidities may be less likely to make it into long-term survivorship.”

-was there any clinical data about treatment available?

A: Unfortunately, no. On p. 13 we state that “No information was available regarding the stage of disease at the time of diagnosis or the treatments received”. It would be important to study the effect of comorbidities in the context of disease stage and type of treatment, as these factors have also been found to influence depression risk in cancer survivors. We are currently planning a study in a large cohort of cancer survivors with detailed information on comorbidities from a population-based cancer registry (from the “European High-Resolution Studies on Cancer”), so we are hopefully going to address this issue soon. However, for the current research it remains a limitation.
